# Preliminary validation of the 15-item WHO ageism experiences scale in a mixed-age UK sample

Aja Louise Murray *, Xuefei Li, Tom Booth

Department of Psychology, University of Edinburgh, Edinburgh, United Kingdom

☯ Aja Louise Murray and Xuefei Li contributed equally to this manuscript.
* aja.murray@ed.ac.uk

## Abstract

### Background

Ageism, defined as stereotypes, prejudices and discrimination based on a person's age is prevalent and negatively impacts individuals and society. In 2016, the World Health Organization (WHO) received a mandate from its Member States to lead the global campaign to combat ageism. However, success in combating ageism is critically dependent on the availability of valid and reliable measures of people's experiences of ageism that can capture its multidimensional nature and as it is directed against multiple age groups across diverse country settings. The goal of current study is to present a preliminary psychometric validation of a measure specifically designed to meet this need: the 15-item *WHO Ageism Experiences Scale.* The scale includes items measuring self-directed, interpersonal, and institutional ageism and spans stereotypes, prejudices, and discrimination.

### Method

We collected data from a UK mixed-age adult sample (*N* = 402), and several psychometric properties were tested, including internal consistency reliability, test-retest reliability, and convergent validity.

### Results

Results suggested that the 15-item *WHO Ageism Experiences Scale* has good internal consistency (where relevant) and test-retest reliability. The exploratory factor analysis supported a one-factor model for self-directed ageism items. The scale also proved to have adequate convergent validity.

### Discussion

The results provide sufficient support for the adoption of the scale in this population and motivates its translation and validation in other settings, as well as the evaluation of a more extended set of psychometric properties in future studies.

**Data availability statement:** All relevant data for this study are publicly available from the OSF repository (https://osf.io/jfa53).

**Funding:** Author ALM was supported by the Wellcome Trust (ref: 225364/Z/22/Z). The funders had no role in study design, data collection and analysis, decision to publish, or preparation of the manuscript.

Ageism, defined as stereotypes, prejudices, and discrimination related to age is a prevalent and urgent issue affecting the global community [1]. Directed against older adults, it is associated with wide-ranging outcomes across health, wellbeing, prosperity and longevity, and may contribute to elder abuse [2,3]. In 2016, the World Health Organization (WHO) received a mandate from its Member States to lead a global campaign to combat ageism. Critical to these efforts is the availability of a globally valid and reliable measure of ageism [1]. Such a measure is crucial to identify patterns, predictors, outcomes and the effects of interventions to reduce ageism.

Although a number of valuable measures of ageism have been developed previously [2,4,5], systematic reviews reported that no measure was available that can meet these needs [2,4]. Most crucially, researchers have identified a lack of measures that capture the multi-dimensional nature of ageism experiences, defined as comprising stereotypes, prejudices, and discrimination; and occurring at the intrapersonal, interpersonal, and institutional level [4]. Several additional shortcomings have also been identified, including a lack of comprehensive psychometric testing and cross-cultural validation, with only a handful of measures designed to capture ageism across a diversity of cultural contexts. This is important given that age-related constructs, including ageism itself may vary in the way it is understood and manifested across countries [1]. There are also a lack of measures that can capture ageism against multiple age groups, despite the fact that ageism can affect all age groups, not just older adults [6].

To address these shortcomings, researchers developed a novel extensive pool of ageism items consistent with contemporary multi-dimensional conceptualisations [7]. The process drew on comprehensive reviews of the literature and ageism expert input, both global in scope. The item pool then underwent content validity review by a group of experts sampled from all WHO-defined major world regions, with expertise in ageism, psychometrics or both. This led to further refinements to the item pool, including the elimination of some items (e.g., for a lack of universal applicability across contexts) and modification of others. The current study describes the validation of a 15-item ageism experiences scale selected from this item pool.

## Method

### Participants

Participants were sampled via Prolific (an online platform for recruitment of study participants) using a convenience sampling method between 25 May and 7 December 2023. Two baseline waves were conducted in May and September, followed by two follow-up waves three months later, in September and December, respectively. Inclusion criteria required participants to be adults (age 18+), able to provide informed consent, and based in the UK. Exclusion criteria included identifying as a student or being unable to provide informed consent. The final sample comprised 402 participants (198 female, 199 male) with a mean age of 40.11 ($SD = 11.01$; Min = 20, Max = 65). 89% of participants identified as White, with the remaining participants identifying as Asian/Asian British (4.2%), Black/African/Caribbean/Black British (2.0%), Mixed/Multiple ethnic groups (1.7%), or Other. Most respondents held a

Bachelor's degree or higher, including 50.2% with a Bachelor's degree, 18.4% with a Master's degree or MBA, and 3% with a PhD or MD. In terms of relationship status, participants were primarily never married (50.2%) or married (41.3%). Sample size requirements depend on properties of the items such as factor loadings that cannot be known in advance, therefore, sample size planning for scale validation can be challenging. The sample size for the present study was based on a resource constraint approach in which the number of participants was maximised within the available resources whilst ensuring that it would meet minimal sample size requirements for psychometric validation for scales of its length. For example, researchers conducted a simulation study, recommending a minimum sample size of 300 for the effective validation of 15-item scales [8]. This size is also considered 'good' according to other established psychometric guidelines [9,10]. Therefore, our sample size aligned with those criteria, allowing us to perform thorough psychometric evaluations.

## Ethics

The study was approved by the School of Philosophy, Psychology and Language Sciences Research Ethics Committee (PPLSREC) at the University of Edinburgh (Ref. 282–2223/3). All participants were adults recruited via the online platform Prolific. Before beginning the survey, they were presented with an online information sheet detailing the study purpose, procedures, and rights. Informed consent was obtained electronically: participants indicated their agreement by selecting the consent option and clicking to proceed on the Prolific platform. No minors were included, so parental or guardian consent was not required.

## Measures

### WHO ageism experiences scale

**Item selection.** Items were selected from the WHO ageism item pool developed by [7]. Items selected should provide a good balance of capturing ageism in all its key aspects, whilst minimising participant burden and making it feasible to include in larger scale studies [11], hence a 15-item scale was judged optimal. Our approach prioritised the preservation of content validity [12] over optimising statistical psychometric properties such as reliability because it is critical to retain coverage of all key domains of ageism. Selection driven by maximising reliability indices can result in items that correlate highly due to being similar in content, resulting in a scale that lacks breadth of content [13]

The item selection process was primarily driven by expert judgement aimed at satisfying the criteria for content validity. Content validity was conceptualised as the extent to which the components of the measure appropriately and thoroughly reflect the target construct for a given assessment goal. Specifically, two key features of content validity are relevance and representativeness [14]. The former refers to the appropriateness of an aspect of a scale to the target construct and the latter to the extent to which the facets of the target construct are proportionally represented by items. Thus, for a scale to possess content validity, it should include items covering the full range of possible manifestations both in terms of content and in terms of 'severity' in proportion to the target construct in the population.

Based on these considerations, items were selected to ensure that all core domains were well-represented in the scale, i.e., *interpersonal and self-directed stereotypes, prejudices, and discrimination* and *institutional discrimination* from an 'experiences' perspective [1,7]. Within the subscales mapping to these focal domains, more general items (e.g., items that were not specific to particular to any context, e.g., work, education, romantic relationships, finances) were prioritised for selection given that these are likely to better capture overall levels of exposure to ageism across contexts. This is important given the goal of a scale that can be used across ages and country contexts. Crucially, the scale's content was restricted to negative ageism (i.e., harmful stereotypes, prejudice, and discrimination). This decision was based on evidence showing negative ageism is associated with greater harm and adverse outcomes than positive (benevolent) ageism, thus prioritising the most consequential aspect of the construct. However, a mixture of positively and negatively worded items were selected to help manage the acceptability of the scale. This was based on feedback at earlier stages of expert content validity review

where the risk of an overly negative scale was highlighted [7]. Items were also selected with the aim of capturing a range of 'severities' of ageism from mild or 'everyday' ageism to extreme ageism. This is to help ensure a scale that can measure a wide range of ageism exposure levels well [15,16]. Finally, discrimination items were given proportionally greater representation than stereotypes and prejudice. This reflects the assumption that discriminatory behaviours are the most direct and tangible forms of ageism and, therefore, may have the greatest impact on those experiencing them.

While a full account of the item generation and rigorous multi-stage elimination process is beyond the scope of this preliminary validation, further information is available in the development paper [7]. It involved multiple stages of refinement. Items lacking broad contextual relevance were removed, and others were modified for clarity and precision. The current study presents the validation of a 15-item ageism experiences scale drawn from this finalised pool.

The selected items for the 15-item version (*WHO-Ageism-E15*) are shown in Table 1. The selected items were from the self-stereotypes (2 items), self-prejudice (1 item), self-discrimination (2 items), interpersonal stereotypes (2 items), interpersonal prejudices (1 item), interpersonal discrimination (3 items) and institutional discrimination (3 items). Of these 15-items, we selected 3 (20%) that were positively worded, e.g., '*At my age, my life has plenty of purpose*'.

**PAQ.** The *Perceived Ageism Questionnaire* (PAQ) [17] was used as a measure of convergent validity. It was selected as it includes items measuring both stereotypes and discrimination. It includes 8 items covering both positive and negative ageism, e.g., 'people assume that you are wise and sensible because of your age' and 'people approach you as if you were a child because of your age'. It was developed in the Netherlands and has been translated into English, the version used in the present study. Previous psychometric analyses of the scale have suggested a two-dimensional structure, with high internal consistency reliability for the resulting two subscales (negative: Cronbach's alpha = .75; positive: Cronbach's alpha = .81) and adequate internal consistency reliability overall (overall: Cronbach's alpha = .63). In an independent sample in the same psychometric study, its two-dimensional structure showed good fit to the data in a Confirmatory Factor Analysis (CFA), with the two dimensions not significantly correlated ($r = .08$; $p = .15$). The scores had internal consistency reliability values of .67 for the overall scale; .71 for the positive subscale; and .79 for the negative subscale.

**Table 1. *WHO ageism experiences 15-item scale.***

| Item number | Content domain | Keying | Item |
|---|---|---|---|
| 1 | Self-stereotypes | + | At my age, my life has plenty of purpose |
| 2 | Self-stereotypes | – | I am a burden because of my age |
| 3 | Self-prejudice | – | I am embarrassed of my age |
| 4 | Self-discrimination | – | Due to my age, I limit my participation in discussions even when they are about things that affect me |
| 5 | Self-discrimination | – | There are things I would like to do if I did not consider them inappropriate for my age group |
| 6 | Interpersonal stereotypes | – | Others think that I have nothing valuable to contribute to society because of my age |
| 7 | Interpersonal stereotypes | + | Others think that at my age I am able to make decisions for myself |
| 8 | Interpersonal prejudices | – | Others feel frustrated with me due to my age |
| 9 | Interpersonal prejudices | – | Other people feel uncomfortable around me because of my age |
| 10 | Interpersonal discrimination | – | Due to my age, other people talk to me as if I need things simplified |
| 11 | Interpersonal discrimination | – | Others make decisions for me because of my age |
| 12 | Interpersonal discrimination | – | Due to my age, others make me feel excluded |
| 13 | Institutional discrimination | – | Policies made by the government (e.g., on housing, social security, healthcare) do not meet the needs of people my age |
| 14 | Institutional discrimination | + | People my age are portrayed positively in the media |
| 15 | Institutional discrimination | – | I have been turned down for an opportunity (e.g., a job or volunteering opportunity) that I was qualified for because of my age |

**Sociodemographic variables.** *Age* was calculated based on self-reported year of birth. *Sex* was reported in the categories, 'male', 'female', 'non-binary/ third gender'. *Health status* was reported on a five- point scale from 'very bad' to 'very good'.

## Statistical procedure

**Overview.** The scale development paper [7] argued that some of the ageism experiences items in the WHO-ageism item pool, i.e., the interpersonal and institutional items have formative (or 'causal') rather than reflective (or 'effect') relations with the concepts they measure [18]. This is based on the idea that for interpersonal and institutional ageism indicators a person's overall experiences of ageism do not reflect an internal latent (psychological) trait. Rather, they are better conceptualised as a composite of experiences across different contexts, and in interaction with multiple different individuals and institutions. For example, a person's experience of interpersonal prejudice (e.g., "Other people feel uncomfortable around me because of my age") and their experience of institutional discrimination (e.g., "Policies made by the government do not meet the needs of people my age") are distinct external experiences that form the overall ageism score, and these experiences are not assumed to be necessarily highly correlated. Self-directed ageism, in contrast can be thought of as reflecting a psychological trait of internalised ageism. The items in this domain are considered reflective indicators, meaning they are the effects or manifestations of this single underlying internal trait.

For formative/causal indicators, psychometric measurement models such a factor analysis and item response theory, which assume an underlying latent trait are not appropriate [19,20]. This is because these models are built on the statistical assumption that all indicators are measuring the same latent cause. When indicators *define* the construct rather than *reflect* it (as in the formative model), this fundamental assumption is violated, making tests like internal consistency reliability and factorial validity conceptually unsound for assessing their psychometric quality. However, formative indicators could still be expected to show high test-rest reliability and convergent and criterion validity with other measures. As such, our analyses of the interpersonal and institutional experiences items focus on these properties. Our analyses of self-directed ageism additionally include analyses of the internal structure and internal consistency reliability of the items.

## Exploratory factor analysis (self-directed ageism items)

Though items were designed to reflect three core conceptual domains (stereotypes, prejudices, and discrimination), the factor structure of the self-directed ageism items is not known *a priori*. For example, it is unclear whether indicators from the domains of stereotypes, prejudices and discrimination will better reflect distinct factors, a single underlying factor, both (e.g., in a bifactor model), or an alternative structure. We, therefore, conducted an Exploratory Factor Analysis (EFA) of the self-directed ageism items. Factor retention was guided by Parallel Analysis with Principal Components (PA-PCA), the Minimum Average Partial test (MAP) and visual inspection of a scree plot. Factor solutions for a range of potential optimal solutions were also inspected and substantive and practical issues considered in model selection. For example, factor solutions were compared on whether they made meaningful distinctions and whether they included poorly determined factors (with few saliently loading items, i.e., loading<|.3|). In these and the final solution, an oblimin (oblique) rotation was used and minimum residual (minres) used for factor extraction.

## Internal consistency reliability (self-directed ageism items)

Omega reliability [21] was estimated for the factors in the factor solution indicated for the self-directed ageism items. Values greater than .7 were considered to indicate good internal consistency reliability.

## Correlations with other measures (all ageism items)

While there is no gold standard measure of ageism to compare the new WHO ageism experiences scale against [4], high correlations with other measures of ageism purporting to measure the same concepts can provide indirect support for the

scale. To evaluate convergent validity, a composite ageism experiences score was derived and correlations estimated between this and ageism measured by the PAQ. As noted above, the PAQ measures both positive and negative ageism. Associations with both concepts were estimated. As the PAQ does not include self-directed ageism items, separate associations were estimated for the WHO self-directed and WHO interpersonal/institutional items as well as the association with the overall WHO ageism score. It was hypothesised that all associations would be significant between WHO and PAQ scores; however, the strongest association would be between the WHO interpersonal/institutional score and the PAQ negative ageism score. Given their focus on negative ageism, the associations between the WHO scales and PAQ positive ageism scores were expected to be weaker than for the PAQ negative ageism items.

To provide further illumination on the interpretation of the scores, we also evaluated the associations with age, sex, education level, marital status, and health status. Ageism scores were regressed on each of these variables separately. For age, both linear and quadratic effects were estimated.

### Test-rest reliability (all items)

A subset of the sample ($N=80$ with up to $N=69$ with sufficiently complete data) completed the WHO ageism scale an average of 96 days after its initial administration. Test-retest reliability was estimated as the correlation between ageism composite scores across baseline and follow-up.

## Results

### Item-level descriptive statistics

Item-level descriptive statistics for the 15 ageism items selected as part of the WHO ageism experiences scales are provided in Table 2. The mean composite scores (with reverse keyed items reverse scored) for self-directed ageism,

Table 2. Descriptive statistics for the *WHO ageism experiences* items.

| Item | Content | N | Mean | SD | Min | Max | Skew | Kurtosis |
|---|---|---|---|---|---|---|---|---|
| 1 | At my age, my life has plenty of purpose | 400 | 4.13 | 0.96 | 1.00 | 5.00 | −1.30 | 1.68 |
| 2 | I am a burden because of my age | 392 | 1.31 | 0.62 | 1.00 | 4.00 | 2.25 | 5.29 |
| 3 | I am embarrassed of my age | 396 | 1.61 | 0.97 | 1.00 | 5.00 | 1.63 | 1.84 |
| 4 | Due to my age, I limit my participation in discussions even when they are about things that affect me | 396 | 1.51 | 0.84 | 1.00 | 5.00 | 1.87 | 3.35 |
| 5 | There are things I would like to do if I did not consider them inappropriate for my age group | 392 | 2.16 | 1.23 | 1.00 | 5.00 | 0.73 | −0.67 |
| 6 | Others think that I have nothing valuable to contribute to society because of my age | 391 | 1.59 | 0.89 | 1.00 | 4.00 | 1.47 | 1.17 |
| 7 | Others think that at my age I am able to make decisions for myself | 398 | 4.10 | 1.34 | 1.00 | 5.00 | −1.42 | 0.62 |
| 8 | Others feel frustrated with me due to my age | 393 | 1.47 | 0.82 | 1.00 | 5.00 | 1.91 | 3.22 |
| 9 | Other people feel uncomfortable around me because of my age | 392 | 1.45 | 0.79 | 1.00 | 4.00 | 1.83 | 2.69 |
| 10 | Due to my age, other people talk to me as if I need things simplified | 393 | 1.42 | 0.83 | 1.00 | 5.00 | 2.26 | 4.85 |
| 11 | Others make decisions for me because of my age | 389 | 1.32 | 0.73 | 1.00 | 5.00 | 2.70 | 7.56 |
| 12 | Due to my age, others make me feel excluded | 390 | 1.52 | 0.85 | 1.00 | 5.00 | 1.75 | 2.48 |
| 13 | Policies made by the government (e.g., on housing, social security, healthcare) do not meet the needs of people my age | 379 | 2.85 | 1.40 | 1.00 | 5.00 | −0.01 | −1.33 |
| 14 | People my age are portrayed positively in the media | 389 | 2.94 | 1.07 | 1.00 | 5.00 | −0.09 | −0.57 |
| 15 | I have been turned down for an opportunity (e.g., a job or volunteering opportunity) that I was qualified for because of my age | 356 | 1.70 | 1.06 | 1.00 | 5.00 | 1.43 | 0.98 |

interpersonal and institutional ageism and overall ageism experiences based on the 15-item scale (5 items for self-directed ageism, 10 for interpersonal and institutional ageism) were: 1.67 ($SD = 0.63$), 1.83 ($SD = 0.6$) and 1.76 ($SD = 0.53$), indicating overall low levels of ageism in these domains in the current sample.

The full breakdown of response frequencies (percentages for scores 1–5 and missing data) for each item is provided in Supplementary Table S1 Table, which showed a pattern of responses reflecting low agreement with perceived ageism experiences.

### EFA of self-directed ageism items

The MAP test, PA-PCA and visual inspection of a scree plot suggested only one factor to retain. The one and two factor solutions were compared (see Supplementary Tables S2 and S3 for the solutions) and the patterns of factor loadings suggested that the one-factor solution was optimal. Specifically, while in the one factor solution all items loaded variably but saliently on a common factor, in the two-factor solution the second factor was almost entirely defined by a since item (item 2 with a loading of .96) and the first item did not load saliently on either factor.

### Internal consistency reliability of the self-directed ageism items

Omega reliability for the self-directed ageism items as measures of a general self-directed ageism factor was .76.

### Correlations with PAQ ageism scores

The correlations between WHO-ageism experiences scores and PAQ ageism scores are provided in Table 3 and visualised in Fig 1. The nodes represent the PAQ and WHO ageism scores, with thicker lines between the nodes representing stronger associations. These indicated a moderate correlation between self-directed and institutional/interpersonal ageism. Both were significantly correlated with PAQ negative ageism scores, with institutional/interpersonal ageism exhibiting the stronger correlation. Only self-directed ageism was significantly associated with positive ageism, and the association was modest in magnitude. Overall, the pattern of correlations supports the convergent validity of the WHO ageism scales, with more closely conceptually related scores exhibiting relatively stronger correlations.

### Associations with sociodemographic characteristics

The associations between the ageism scores sociodemographic variables are shown in Table 4. No significant linear or quadratic relationships were found between age and self-directed ageism. However, age was significantly associated with dimensions of interpersonal and institutional ageism (both linear and quadratic effects). Better health was associated with less self-directed and interpersonal/institutional ageism. Education was not significantly associated with either dimension of ageism. Married individuals reported higher self-directed and interpersonal/institutional ageism compared with those who were not married. Sex was not significantly associated with either ageism dimension.

Table 3. Correlation matrix of ageism scores.

| | Self-directed ageism | Institutional and interpersonal ageism | PAQ positive ageism | PAQ negative ageism |
|---|---|---|---|---|
| **Self-directed ageism** | – | <.001 | .19 | <.001 |
| **Institutional and interpersonal ageism** | .58* | – | <.001 | <.001 |
| **PAQ positive ageism** | −.07 | −.16* | – | <.001 |
| **PAQ negative ageism** | .35* | .63* | −.06 | – |

*Note.* Pearson correlations below the diagonal and *p*-values above.

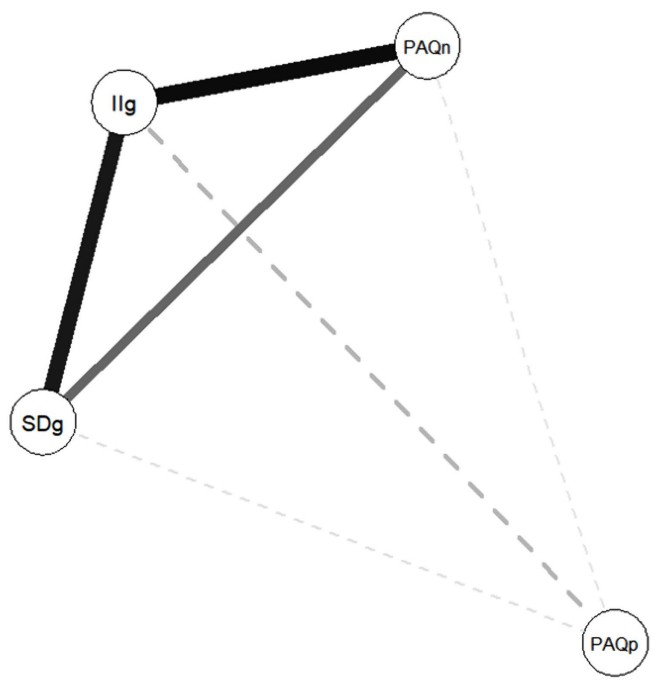

**Fig 1. Visualisation of ageism score patterns.** *Note.* SDg = Self-directed ageism, IIg = Interpersonal and institutional ageism, PAQn = PAQ negative ageism, PAQp = PAQ positive ageism.

**Table 4. Associations between ageism scores and sociodemographic variables.**

| Variable | B Self-directed ageism | p | B interpersonal/institutional ageism | p |
|---|---|---|---|---|
| Age | −0.002 | .92 | −.12 | <.001 |
| Age^2 | 0.000 | .74 | .001 | <.001 |
| Sex (reference category = female) | 0.036 | .59 | −0.029 | .66 |
| Education level | −0.063 | 0.134 | −0.014 | .71 |
| Marital status (reference = not married) | 0.135 | **0.011** | 0.158 | **<.001** |
| Health status | −0.25 | **<.001** | −0.20 | **<.001** |

*Note.* Only male and female participants were included in the sex-based analyses due to insufficient representation of other sexes/genders to permit statistically powered comparisons. Education level was treated as a continuous variable, with higher values indicating higher educational attainment. For marital status, individuals who were not married served as the reference group to enable adequately powered comparisons with those who were married.

## Test-retest reliability

Among the participants with ageism data across time, test-retest correlations were for self-directed ageism $r = .71$ ($N = 69$, $p < .001$); interpersonal/institutional ageism $r = .63$, ($N = 59$, $p < .001$); and general ageism $r = .72$ ($N = 53$, $p < .011$).

## Discussion

The purpose of the present study was to provide an initial psychometric evaluation of the newly developed 15 item WHO Ageism Experiences Scale. It was developed to address a lack of suitable measures that could be used to assess ageism experiences in all its key dimensions, against all ages, and across the globe [7]. Our analysis of data gathered from

a mixed-age adult UK sample provided support for the scale scores. An EFA of the self-directed ageism items (the 'effect indicators') suggested a one-factor model, with high internal consistency reliability. Convergent validity and test-retest reliability for these and the interpersonal/institutional items (which were assumed to be 'causal indicators') were also supported. Ageism scores were associated with poorer health but not with age or sex.

The evaluation of the WHO ageism experiences scale is complicated in comprising a mix of formative and reflective (or 'cause' and 'effect') indicators. In contrast to reflective indicators which are assumed to 'reflect' an underlying latent construct and therefore covary as a result of their co-dependence on it, formative indicators combine to define a construct [18]. Most tests are developed on a reflective variable assumption; however, there are a number of constructs that have been previously discussed as formative. Socioeconomic status (SES) is often cited as an example of being composed of or 'constructed from' formative indicators such as occupational prestige, income, and educational level [22]. In this sense, the indicators define the construct and there is no assumption of a latent causal SES factor. Other constructs that have been proposed to involve formative indicator-construct relations include job satisfaction, exposure to life stress, and degree of social interaction [20,23].

Analogous considerations when applied to ageism indicators suggest that self-directed ageism could be conceptualised as a latent psychological trait that determines specific self-directed ageist cognitions, emotions, and behaviours, making these indicator-construct relations reflective. However, a formative interpretation is arguably more suitable for the interpersonal and institutional ageism items as ageism experiences from the external environment can combine from heterogeneous sources, namely, a diversity of individuals and institutions, rather than a single unitary factor. For more details on the conceptual and statistical distinction between formative and reflective indicators, please refer to [18,19,24,25].

In addition to the core psychometric findings of the present study, a further notable finding was the lack of association between positive and negative aspects of ageism. The finding that the WHO scale (which measures negative ageism) was strongly correlated with the PAQ negative ageism subscale but only weakly with the PAQ positive ageism subscale supports the convergent validity of our measure. This pattern also provides empirical support for the theoretical distinction between negative and positive ageism, suggesting they are distinct constructs. It has been noted that it is important to measure positive ageism (e.g., stereotypes that older adults are wise) [17,26]. In the early stages of the WHO ageism scales conceptualisation there were thus discussions around whether the scale should measure both positive and negative aspects of ageism or just the latter. It was concluded that since the scales would need to cover a wide range of content and that negative ageism is associated with greater harm, the focus would be limited to negative ageism. The present results arguably provide further support for this decision from a different perspective. Specifically, they suggest that negative ageism can be treated as a distinct construct from positive ageism. Positive ageism is relatively under-researched and a valuable future direction for research would be to explore how positive ageism, alongside negative ageism, impacts individuals of different age groups [27].

Finally, while detailed analysis of item response distributions suggested that most participants selected the response category indicating lowest ageism level, this pattern is highly consistent with our expectations given the nature of the current sample and alignment with existing literature. Specifically, the sample is a mixed-age sample, which is not considered a high-risk group for ageism experiences, and the observed low means accurately reflect the low prevalence of explicit ageism experiences in this population. Prior literature on age discrimination measures in large-scale, population-based surveys like English Longitudinal Study of Ageing (ELSA) and Health and Retirement Study (HRS) [28,29] has consistently shown that responses are highly skewed towards the "never experienced" category, even in older-adult cohorts. Thus, we concluded that the observed distribution is a representation of the phenomenon in this population.

### Strengths and limitations

It is important to note the limitations of the present study. First, our sample was a non-probability sample and is not representative of the wider UK population. Crucially, while the sample was drawn from the UK, it was largely ethnically homogeneous (primarily White British), which restricts the generalisability of the findings across diverse ethnic subgroups.

Similarly, though different age groups may have different experiences of ageism, we didn't have sufficient numbers of participants to conduct age-stratified analyses, as well as an underrepresentation of participants in the older age range of the sample (i.e., those aged 50–65). The positive findings of the present study support the use of WHO ageism experiences scale for use in UK mixed-age adult samples. However, further work will be required to extend the validation to other populations; including the adaptation and translation of the scale for use in a wider age range, a diversity of additional global contexts and ethnically diverse groups. Future validation studies in populations where ageism is expected to be more prevalent (e.g., older adult clinical samples or specific cultural setting) will be essential to test the scale's full range sensitivity. It is also worth noting that the sample in this study comprised adults across a wide age range. Although several studies have validated this scale among older adults [30,31] and reported findings consistent with ours, generalisability to specific populations such as older adults should still be interpreted with caution. Further, given the lack of a priori evidence on the optimal factor structure for the self-directed ageism items, we took an EFA approach in the present study. Future research will be required to test whether this can be confirmed via confirmatory factor analysis in future, independent samples. Moreover, another limitation of the current validation study is the absence of an external measure specifically designed to capture internalised or self-directed ageism for convergent validation. While the PAQ was included for its ability to measure both stereotypes and discrimination, and its coverage of both negative and positive ageism, it does not contain items targeting the self-directed dimension. This restricts the ability to fully establish the convergent validity of the self-directed ageism subscale. Future research should incorporate an established external measure of internalised ageism to more rigorously evaluate the convergent validity of this specific subscale. One suitable option is the Aging Perceptions Questionnaire (APQ) [32], which assesses individuals' beliefs and feelings about their own ageing and has been widely used as a measure of self-directed or internalised ageism. The APQ also offers shorter validated versions (e.g., the Brief APQ [B-APQ] and the Short APQ [APQ-S]), making it feasible to include in future studies alongside the current scale. Finally, there were several psychometric properties which we did not test in the current study due to data limitations, which could be assessed in the future, including correlations with a wider range of criterion variables, investigation of discriminant validity with measures of distinct constructs (e.g., general negative affect), measurement invariance across core groups (e.g., age, gender, ethnicity, and health status); and sensitivity to interventions [33]. Similarly, while content validity has been assessed by experts in a previous study, cognitive interviews would be helpful to further explore how the items are understood participants [34].

## Conclusions

Initial evidence supports the psychometric properties of the WHO Ageism Experiences Scale 15-item scale in a mixed-age UK sample. Future research will be needed to provide further validation in a wider range of psychometric properties and across diverse settings, languages and population.

## Supporting information

**S1 Table. Percentage distribution of responses across ageism items.**
(DOCX)

**S2 Table. Factor loadings for self-directed ageism one-factor solution.**
(DOCX)

**S3 Table. Factor loadings for self-directed ageism two-factor solution.**
(DOCX)

## Acknowledgments

We are grateful to Dr. Christopher Mikton for his invaluable input on this manuscript.

## Author contributions

**Conceptualization:** Aja Louise Murray, Xuefei Li, Tom Booth.

**Formal analysis:** Aja Louise Murray.

**Methodology:** Aja Louise Murray, Xuefei Li, Tom Booth.

**Project administration:** Aja Louise Murray, Tom Booth.

**Validation:** Aja Louise Murray, Xuefei Li.

**Visualization:** Aja Louise Murray.

**Writing – original draft:** Aja Louise Murray, Xuefei Li.

**Writing – review & editing:** Aja Louise Murray, Xuefei Li, Tom Booth.

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
