## [Decision Letter · Decision Letter 0]

29 Oct 2025

PONE-D-25-50563Preliminary validation of the 15-item WHO ageism experiences scale in a mixed-age UK samplePLOS ONE

Dear Dr. Murray,

Thank you for submitting your manuscript to PLOS ONE. After careful consideration, we feel that it has merit but does not fully meet PLOS ONE’s publication criteria as it currently stands. Therefore, we invite you to submit a revised version of the manuscript that considers the points raised by the reviewers that are listed below.

If applicable, we recommend that you deposit your laboratory protocols in protocols.io to enhance the reproducibility of your results. Protocols.io assigns your protocol its own identifier (DOI) so that it can be cited independently in the future. For instructions see: https://journals.plos.org/plosone/s/submission-guidelines#loc-laboratory-protocols. Additionally, PLOS ONE offers an option for publishing peer-reviewed Lab Protocol articles, which describe protocols hosted on protocols.io. Read more information on sharing protocols at . Additionally, PLOS ONE offers an option for publishing peer-reviewed Lab Protocol articles, which describe protocols hosted on protocols.io. Read more information on sharing protocols at https://plos.org/protocols?utm_medium=editorial-email&utm_source=authorletters&utm_campaign=protocols..

We look forward to receiving your revised manuscript.

Kind regards,

Antony Bayer

Academic Editor

PLOS ONE

Journal Requirements:

“Author ALM was supported by the Wellcome Trust (ref: 225364/Z/22/Z). ”

Please include this amended Role of Funder statement in your cover letter; we will change the online submission form on your behalf

3. Please include captions for your Supporting Information files at the end of your manuscript, and update any in-text citations to match accordingly. Please see our Supporting Information guidelines for more information: http://journals.plos.org/plosone/s/supporting-information..

Reviewers' comments:

Reviewer's Responses to Questions

**Comments to the Author**

1. Is the manuscript technically sound, and do the data support the conclusions?

Reviewer #1: Yes

Reviewer #2: Yes

2. Has the statistical analysis been performed appropriately and rigorously? 

Reviewer #1: Yes

Reviewer #2: Yes

3. Have the authors made all data underlying the findings in their manuscript fully available?

Reviewer #1: Yes

Reviewer #2: No

4. Is the manuscript presented in an intelligible fashion and written in standard English?

Reviewer #1: Yes

Reviewer #2: Yes

5. Review Comments to the Author

Reviewer #1: This paper presents a preliminary validation of the 15-item WHO ageism experiences scale in a mixed-age UK sample, aiming to address a documented gap in the availability of psychometrically sound measures of ageism. The study's focus on capturing the multidimensional nature of ageism across different age groups and contexts is potentially valuable, considering limitations in the existing measurement landscape.

Strengths:

• Relevance and Timeliness: The research aligns with the WHO's global campaign to combat ageism and responds to the urgent need for reliable tools to measure ageism experiences.

• Clear Conceptual Framework: The paper defines ageism and its dimensions (self-directed, interpersonal, institutional) providing a rationale for the scale's development.

• Sound Methodology: The use of exploratory factor analysis, internal consistency, test-retest reliability, and convergent validity offers an initial evaluation of the scale's properties.

• Promising Initial Findings: The results suggest initial good internal consistency and test-retest reliability.

Areas for Improvement (Critical Concerns):

• Sample Size and Selection: The justification for the sample size (N=402) relies on resource constraints. While this is understandable, there's a lack of clear explanation concerning the adequacy of the sample size. In addition, elaboration on the selection criteria of the sample is needed. What are the specific inclusion/exclusion criteria?

• Item Selection and Justification: The paper starts with 15 items and concludes with 15 items. The stability raises concern about content validity. The selection criteria for the scale's 15 items from the WHO ageism item pool are inadequately justified. The rationale for why these specific 15 items were chosen to cover all relevant aspects of ageism needs to be transparent. It is not clear if it sufficiently captured key aspects of the construct of interest, raising doubts about content validity.

• Construct Validity Concerns: The absence of a measure of self-directed ageism for construct validation is an oversight. Including such a measure would have provided a stronger test of whether the scale is indeed measuring ageism experiences.

• Deeper Dive into Construct Validity: While convergent validity is assessed, further investigation into discriminant validity would strengthen the argument that the scale measures ageism specifically, and not other related constructs.

• Exploring Subgroup Differences: The authors acknowledge the lack of age-stratified analyses. However, exploring potential differences in ageism experiences based on other sociodemographic variables (e.g., gender, ethnicity) could provide valuable insights into the scale's performance across diverse subgroups.

• Clarity on Formative vs. Reflective: The paper's discussion of formative vs. reflective indicators requires further clarification. The statement "For formative/causal indicators, psychometric measurement models such as a factor analysis and item response theory, which assume an underlying latent trait are not appropriate" needs to be explained more thoroughly. A more accessible explanation of the underlying statistical assumptions and why they are violated in the case of formative indicators is necessary.

• Factor Analysis Limitations: The authors did not attempt to conduct Factor Analysis on interpersonal and institutional ageism. The explanation for why it’s not possible remains somewhat unclear. If there is a good reason, it has to be clearly explained so that the readers can fully understand the decision.

• Dimensionality of Ageism: The authors should have explained why they didn’t attempt to conduct FA to see whether the 3 dimensions of ageism can be teased apart- e.g., intra, inter, and institutional ageism. Addressing the scale's ability to differentiate these dimensions is essential for its usefulness in understanding and addressing ageism.

• Floor and Ceiling Effects: The paper does not address potential floor or ceiling effects. Given that the study indicates "overall low levels of ageism," it is critical to examine whether the scale is sensitive enough to detect ageism experiences across the entire spectrum.

• Discriminant Validity: The omission of an assessment of discriminant validity is a significant weakness. Demonstrating that the scale measures something distinct from related constructs (e.g., general negativity, depression) is essential for establishing its unique contribution.

• Justification for Formative Interpretation: "However, a formative interpretation is arguably more suitable for the interpersonal and institutional ageism items as ageism experiences from the external environment can combine from heterogeneous sources, namely, a diversity of individuals and institutions, rather than a single unitary factor”. There are no clear reasons or more detail description to fully support the claim.

• Positive vs. Negative Ageism: The decision to focus on negative ageism is unclear. The abrupt introduction of the positive vs. negative ageism in the discussion section, without previous reference or link to the measure's development is unclear. It's essential to justify how this decision informed the scale's content and design.

Editorial Errors:

• The word "was" should be deleted on page 12 "correlations was are provided in Table 3".

Reviewer #2: In this paper, the authors describe the development of an experiences of ageism scale. In general, the development of the measure is well-conceived and thorough. There are several spots in the manuscript, however, where the authors could provide more information and improve clarity. Please see more detailed comments below.

1. Abstract: did the authors test concurrent or convergent validity? They used both terms in the method and results section.

2. Introduction and method. On p. 4 and 5-7 the authors describe the process of developing an item pool and then winnowing it down to the 15 items tested for this paper. More detail should be provided about the larger overall item pool and how the item elimination process proceeded.

3. Sample. The authors collected their data using the platform Prolific. More demographic information should be provided about the sample, including the age range, education levels, income levels, countries of origin, self-rated health, etc.

Knowing the age range and also the breakdown of the number of young adult, middle-aged adult, and older adult participants is important, given the authors have the goal of developing a measure that can be used for multiple age groups. Given the mean age is 40, it appears that middle-aged adults (and likely younger adults) are overrepresented in the sample, and that older adults might be underrepresented.

4. It is clear that the authors consulted and put a lot of thought into the development of the item pool and eventual selection of the 15 items. They note that they made an effort to include some positively worded items, however, only 3 of the 15 are phrased positively. Isn’t it problematic that some of the types of ageism items they included were only captured using negative phrasing? Why weren’t at least 5 out of 15 items worded positively, so that a response set could be prevented as well as avoiding constructing an overly negative scale?

5. The rationale the authors provide for why internal consistency is not calculated for all of the different subscales of the measure is confusing (p. 8). I have never encountered the arguments they are making for why some psychometric statistics are calculated for some items but not for others.

6. Did the authors collect education levels, income levels, employment status, and country of origin? It would be helpful to know if any of those sociodemographic factors correlated with the ageism measure, given health status does.

7. Discussion. The authors indicate this is a ‘mixed age UK sample’ but they do not specify that in their description of collecting data via the platform Prolific. They should be much clearer on where their sample was recruited from and the sociodemographic of the sample.

6. PLOS authors have the option to publish the peer review history of their article (what does this mean?). If published, this will include your full peer review and any attached files.). If published, this will include your full peer review and any attached files.

.

Reviewer #1: No

Reviewer #2: No

---

## [Author Response · Author response to Decision Letter 1]

30 Jan 2026

PONE-D-25-50563

Response to Reviewers

Dear Dr. Antony Bayer,

Thanks for giving us the opportunity to submit a revised draft of the manuscript “Preliminary validation of the 15-item WHO ageism experiences scale in a mixed-age UK sample” for publication in PLOS One. We appreciate the time and effort that you and the reviewers dedicated to providing feedback on our manuscript and are grateful for the insightful comments on and valuable improvements to our paper.

We have incorporated suggestions made by the reviewers. Those changes are highlighted within the manuscript. Please see below, in blue, for a point-by-point response to the reviewers’ comments and concerns, and quoted text from the manuscripts is in purple.

Reviewers' Comments to the Authors:

Reviewer #1:

This paper presents a preliminary validation of the 15-item WHO ageism experiences scale in a mixed-age UK sample, aiming to address a documented gap in the availability of psychometrically sound measures of ageism. The study's focus on capturing the multidimensional nature of ageism across different age groups and contexts is potentially valuable, considering limitations in the existing measurement landscape.

Response: We thank the reviewer for the careful reading of the manuscript and for the positive assessment of the study's goal. We believe that the development of a valid and reliable measure capable of capturing the multidimensional nature of ageism, as called for by the WHO, is critical to advancing global efforts to combat this issue. We hope this preliminary validation study serves as an important first step in introducing the 15-item WHO ageism experiences scale to a wider audience and providing the necessary initial psychometric evidence to foster confidence in its adoption for future research.

Strengths:

• Relevance and Timeliness: The research aligns with the WHO's global campaign to combat ageism and responds to the urgent need for reliable tools to measure ageism experiences.

• Clear Conceptual Framework: The paper defines ageism and its dimensions (self-directed, interpersonal, institutional) providing a rationale for the scale's development.

• Sound Methodology: The use of exploratory factor analysis, internal consistency, test-retest reliability, and convergent validity offers an initial evaluation of the scale's properties.

• Promising Initial Findings: The results suggest initial good internal consistency and test-retest reliability.

Response: We appreciate the reviewer highlighting the relevance and timeliness of this research, particularly its alignment with the WHO mandate to combat ageism, and its clear conceptual framework. We are encouraged that the reviewer acknowledges the sound methodology employed and agrees that the promising initial findings support the scale's internal consistency, test-retest reliability, and potential for broader use.

Areas for Improvement (Critical Concerns):

Comments1-1: Sample Size and Selection: The justification for the sample size (N=402) relies on resource constraints. While this is understandable, there's a lack of clear explanation concerning the adequacy of the sample size. In addition, elaboration on the selection criteria of the sample is needed. What are the specific inclusion/exclusion criteria?

Response: We thank the reviewer for raising these critical points regarding sample size and participant selection.

1. Sample size and adequacy: We acknowledge that our sample size was influenced by resource constraints, and we fully agree that a larger sample would have been preferable. However, the achieved sample size of n = 402 meets and exceeds several widely cited recommendations for factor analysis and scale validation. Reviews on sample size requirements for psychometric validation (Anthoine et al., 2014; Gunawan et al., 2021) suggested that although guidelines vary across studies, there is broad agreement that sample sizes of 50 are considered “very poor,” 100 “poor,” 200 “fair,” 300 “good,” 500 “very good,” and ≥1,000 “excellent”. Moreover, a complementary rule-of-thumb is the item-to-response ratio (p:N), with recommended ratios ranging between 1:3 and 1:20. Given that our scale contains 15 items and our sample includes 402 participants, our ratio is approximately 1:26.8, which is above the recommended ranges. To clarify this in the manuscript, we have added the following text: “The sample size requirements depend on properties of the items such as factor loadings that cannot be known in advance, therefore, sample size planning for scale validation can be challenging. The sample size for the present study was based on a resource constraint approach in which the number of participants was maximised within the available resources whilst ensuring that it would meet minimal sample size requirements for psychometric validation for scales of its length. For example, Rouquette and Falissard (2011) conducted a simulation study, recommending a minimum sample size of 300 for the effective validation of 15-item scales. This size is also considered 'good' according to other established psychometric guidelines (Anthoine et al., 2014; Gunawan et al., 2021). Therefore, our sample size aligned with those criteria, allowing us to perform thorough psychometric evaluations.”

2. Selection criteria: We agree with the need for greater clarity on inclusion/exclusion criteria. We clarify that participants were recruited via the Prolific platform using convenience sampling. Inclusion criteria were: being an adult (age 18+), residing in the UK, and being able to provide informed consent. Exclusion criteria were: identifying as a student, and being unable to provide informed consent. We also aimed to achieve a gender-balanced sample, which resulted in 198 female and 199 male participants being included. We have updated the main text to reflect these specific criteria more explicitly. The revised text reads as follows on Method-participants section. “Inclusion criteria required participants to be adults (age 18+), able to provide informed consent, and based in the UK. Exclusion criteria included identifying as a student or being unable to provide informed consent.”

Comments1-2: Item Selection and Justification: The paper starts with 15 items and concludes with 15 items. The stability raises concern about content validity. The selection criteria for the scale's 15 items from the WHO ageism item pool are inadequately justified. The rationale for why these specific 15 items were chosen to cover all relevant aspects of ageism needs to be transparent. It is not clear if it sufficiently captured key aspects of the construct of interest, raising doubts about content validity.

Response: We appreciate the reviewer's concern regarding the transparency of the 15-item selection process and its relationship to content validity. We clarify that the final 15-item scale was selected prior to this validation study from a comprehensive item pool developed by Murray and de la Fuente-Núñez (2023). The selection process was driven by a commitment to maximising content validity, prioritising both relevance and representativeness of the core ageism domains. Following the framework of(Haynes et al., 1995), we defined content validity as ‘the degree to which the elements of an assessment are relevant to and representative of the targeted construct for a particular assessment purpose’. Item selection was, therefore, driven by expert judgement regarding how well specific items satisfied these two criteria relative to others in the comprehensive item pool. Specifically:

- Cover all key domains align with Levy (2009)’s definition and the Global report on Ageism from WHO. Specifically, it spanned three components (stereotypes, prejudices, and discrimination), at three levels (self-directed interpersonal, and institutional level). To our knowledge, this is the first scale that can adequately capture all three aspects with respect to ageism experiences.

- Prioritise general items to capture overall exposure across contexts, making the scale feasible for use across diverse ages groups and countries.

- Balance the scale, including a mixture of positively and negatively worded items to manage the risk of an overly negative scale (which happens in many other scales about age discrimination, or age stereotypes)

- Capture a rage of severity range, from everyday to more extreme.

- Give proportionally greater representation to discrimination items to reflect the assumption that the behaviour aspect is the most direct and impactful form of ageism.

To further increase transparency regarding the selection process and content validity considerations, we have added a discussion of the specific criteria (relevance and representativeness) to the Method section of the manuscript. Additionally, we have included a brief discussion of items that were not selected (e.g., 'Others feel anger towards me because of my age' was excluded as it was deemed conceptually redundant or less optimal than a similar, less extreme item) and have indicated that the full non-selected item pool is available in the supplemental materials of the item development paper (Murray & De La Fuente-Núñez, 2023).

The following texts were added to the main texts: “The item selection process was primarily driven by expert judgement aimed at satisfying the criteria for content validity. Content validity was conceptualised as the extent to which the components of the measure appropriately and thoroughly reflect the target construct for a given assessment goal. Specifically, two key features of content validity are relevance and representativeness (Haynes et al., 1995). The former refers to the appropriateness of an aspect of a scale to the target construct and the latter to the extent to which the facets of the target construct are proportionally represented by items. Thus, for a scale to possess content validity, it should include items covering the full range of possible manifestations both in terms of specific types of symptoms and in terms of ‘severity’ in proportion to the target construct in the population.”

Comments1-3: Construct Validity Concerns: The absence of a measure of self-directed ageism for construct validation is an oversight. Including such a measure would have provided a stronger test of whether the scale is indeed measuring ageism experiences.

Response: We thank the reviewer for this insightful comment regarding the construct validity of the self-directed ageism component. We acknowledge that the absence of an external measure specifically assessing self-directed or internalised ageism is a limitation of the present study and restricts our ability to fully establish the convergent validity of this subscale. The Perceived Ageism Questionnaire (PAQ) was selected because it captures both stereotypes and discrimination, and includes both negative and positive ageism, making it a useful comparator for the interpersonal and institutional dimensions. However, as highlighted in a recent review of ageism experience measures (Murray & Li, 2025), no existing instrument, except the WHO Ageism Experiences Scale, adequately covers all levels and dimensions of ageism experiences.

We have now added this point to the limitations section of the manuscript and suggested that future research should incorporate an external measure specifically targeting self-directed ageism to more rigorously evaluate the convergent validity of this subscale. In Limitation section, we added: “Moreover, another limitation of the current validation study is the absence of an external measure specifically designed to capture internalised or self-directed ageism for convergent validation. While the Perceived Ageism Questionnaire (PAQ) was included for its ability to measure both stereotypes and discrimination, and its coverage of both negative and positive ageism, it does not contain items targeting the self-directed dimension. This restricts the ability to fully establish the convergent validity of the self-directed ageism subscale”. We also added “Future research should incorporate an established external measure of internalised ageism to more rigorously evaluate the convergent validity of this specific subscale. One suitable option is the Aging Perceptions Questionnaire (APQ; Barker et al., 2007), which assesses individuals’ beliefs and feelings about their own ageing and has been widely used as a measure of self-directed or internalised ageism. The APQ also offers shorter validated versions (e.g., the Brief APQ [B-APQ] and the Short APQ [APQ-S]), making it feasible to include in future studies alongside the current scale.”

Despite this limitation, several indicators support the internal validity of the self-directed ageism subscale. The exploratory factor analysis yielded a clear one-factor solution for the self-directed items, suggesting that they coherently represent a single underlying construct. In addition, the subscale demonstrated good internal consistency and strong test-retest reliability, providing further evidence that the items reliably measure the intended construct.

Comments1-4: Deeper Dive into Construct Validity: While convergent validity is assessed, further investigation into discriminant validity would strengthen the argument that the scale measures ageism specifically, and not other related constructs.

Response: We thank the reviewer for this comment. We agree with the reviewer that further investigation into discriminant validity would significantly strengthen the argument for the scale's construct validity, demonstrating that it measures ageism experiences distinct from other psychological constructs. While the current study focused on convergent validity using the PAQ, and provided some initial interpretive illumination by assessing associations with sociodemographics, we did not include a measure of a theoretically distinct construct (e.g., general negative affect) to rigorously test discriminant validity. We recognise this as an important area for improvement. We have added a statement to the Discussion section, under Strengths and Limitations, to underscore the need for future studies to incorporate measures that allow for a robust assessment of discriminant validity. The added text reads as follows: “Finally, there were several psychometric properties which we did not test in the current study due to data limitations, which could be assessed in the future, including correlations with a wider range of criterion variables, investigation of discriminant validity with measures of distinct constructs (e.g., general negative affect), measurement invariance across core groups (e.g., age, gender, and health status); and sensitivity to interventions (Mokkink et al., 2010).”

Comments1-5: Exploring Subgroup Differences: The authors acknowledge the lack of age-stratified analyses. However, exploring potential differences in ageism experiences based on other sociodemographic variables (e.g., gender, ethnicity) could provide valuable insights into the scale's performance across diverse subgroups.

Response: We thank the reviewer for highlighting the importance of exploring potential differences in ageism experiences across various sociodemographic subgroups.

We acknowledge the limitations of our preliminary validation study: 1) While we employed a mixed-age UK adult sample, we recognise the sample was largely ethnically homogeneous (primarily White British). This restricts the immediate generalisability of the findings across diverse ethnic groups within the UK and globally; 2) Due to insufficient numbers in specific age bands, we could not conduct age-stratified analyses. However, we have included initial analyses on subgroup differences, as presented in Table 4, which showed that the association between age (and age2), sex, and health status, with both the self-directed and interpersonal/institutional domains.

We agree that exploring additional variables (such as ethnicity) and conducting robust testing for measurement invariance is crucial for future work to ensure the scale performs consistently across diverse subgroups. We have updated the Limitations section to explicitly address the ethnic homogeneity and reinforce the need for future testing across age, gender, ethnicity, and health

---

## [Decision Letter · Decision Letter 1]

23 Feb 2026

PONE-D-25-50563R1Preliminary validation of the 15-item WHO ageism experiences scale in a mixed-age UK samplePLOS One

Dear Dr. Murray,

Thank you for submitting your revised manuscript to PLOS ONE and for your detailed attention to the issues raised by the reviewers. After further consideration, we feel that it has merit but does not fully meet PLOS ONE’s publication criteria as it currently stands. The references need attention so that they are consistent and complete and in keeping with PLOS ONE style (Vancouver/NLM, including all authors, recognised journal title abbreviations, doi, etc.), numbered in text rather than author names and without duplicates (1 and 4). Abbreviations should be spelt out when they first occur. Also, on page 16 you refer to “older adults” and given that your upper age limit was 65, it would be helpful to clarify what age group is meant.We invite you to submit a revised version of the manuscript that addresses these issues.  Please submit your revised manuscript by Apr 09 2026 11:59PM. If you will need more time than this to complete your revisions, please reply to this message or contact the journal office at plosone@plos.org. . Please include the following items when submitting your revised manuscript:

If applicable, we recommend that you deposit your laboratory protocols in protocols.io to enhance the reproducibility of your results. Protocols.io assigns your protocol its own identifier (DOI) so that it can be cited independently in the future. For instructions see: https://journals.plos.org/plosone/s/submission-guidelines#loc-laboratory-protocols. Additionally, PLOS ONE offers an option for publishing peer-reviewed Lab Protocol articles, which describe protocols hosted on protocols.io. Read more information on sharing protocols at . Additionally, PLOS ONE offers an option for publishing peer-reviewed Lab Protocol articles, which describe protocols hosted on protocols.io. Read more information on sharing protocols at https://plos.org/protocols?utm_medium=editorial-email&utm_source=authorletters&utm_campaign=protocols..

We look forward to receiving your revised manuscript.

Kind regards,

Antony Bayer

Academic Editor

PLOS One

Journal Requirements:

Reviewers' comments:

Reviewer's Responses to Questions

**Comments to the Author**

1. If the authors have adequately addressed your comments raised in a previous round of review and you feel that this manuscript is now acceptable for publication, you may indicate that here to bypass the “Comments to the Author” section, enter your conflict of interest statement in the “Confidential to Editor” section, and submit your "Accept" recommendation.

Reviewer #2: All comments have been addressed

2. Is the manuscript technically sound, and do the data support the conclusions?

Reviewer #2: (No Response)

3. Has the statistical analysis been performed appropriately and rigorously? 

Reviewer #2: (No Response)

4. Have the authors made all data underlying the findings in their manuscript fully available?

Reviewer #2: (No Response)

5. Is the manuscript presented in an intelligible fashion and written in standard English?

Reviewer #2: (No Response)

6. Review Comments to the Author

Reviewer #2: (No Response)

7. PLOS authors have the option to publish the peer review history of their article (what does this mean?). If published, this will include your full peer review and any attached files.). If published, this will include your full peer review and any attached files.

.

Reviewer #2: No

---

## [Author Response · Author response to Decision Letter 2]

25 Mar 2026

PONE-D-25-50563

Response to Reviewers

Dear Dr. Antony Bayer,

Thanks for giving us the opportunity to submit a revised draft of the manuscript “Preliminary validation of the 15-item WHO ageism experiences scale in a mixed-age UK sample” for publication in PLOS One. We appreciate the time and effort that you and the reviewers dedicated to providing feedback on our manuscript and are grateful for the insightful comments on and valuable improvements to our paper.

We have incorporated suggestions made by the reviewers. Those changes are highlighted within the manuscript. Please see below, in blue, for a point-by-point response to the reviewers’ comments and concerns, and quoted text from the manuscripts is in purple.

Comments from Editor and reviewer:

1. The references need attention so that they are consistent and complete and in keeping with PLOS ONE style (Vancouver/NLM, including all authors, recognised journal title abbreviations, doi, etc.), numbered in text rather than author names and without duplicates (1 and 4).

Author Response: We thank the reviewer for this observation. We have thoroughly revised the reference list to comply with PLOS ONE Vancouver/NLM style. Specifically:

• References 1 and 4 were identical (World Health Organization. Global report on ageism. 2021). We have removed the duplicate entry and renumbered all in-text citations accordingly.

• All references now include full author lists, recognised NLM journal title abbreviations, and DOIs where available.

• All in-text citations have been converted from author-date format to Vancouver numbered style throughout the manuscript.

• We have verified that all references cited in the text appear in the reference list and vice versa.

2. Abbreviations should be spelt out when they first occur.

Author Response: We have reviewed the full manuscript and ensured that all abbreviations are defined at first use. Abbreviations now introduced in full on first occurrence include: World Health Organization (WHO), exploratory factor analysis (EFA), parallel analysis with principal components (PA-PCA), minimum average partial test (MAP), socioeconomic status (SES), English Longitudinal Study of Ageing (ELSA), Health and Retirement Study (HRS), and Aging Perceptions Questionnaire (APQ).

3. Also, on page 16 you refer to “older adults” and given that your upper age limit was 65, it would be helpful to clarify what age group is meant.

Author Response: We thank the reviewer for highlighting this imprecision. We have revised the sentence to read: "…as well as an underrepresentation of participants in the older age range of the sample (i.e., those aged 50–65)." This more accurately reflects the composition of our sample, in which the maximum age was 65 and no participants above this threshold were included.

---

## [Editor Report · Decision Letter 2]

26 Mar 2026

Preliminary validation of the 15-item WHO ageism experiences scale in a mixed-age UK sample

PONE-D-25-50563R2

Dear Dr. Murray,

We’re pleased to inform you that your manuscript has been judged scientifically suitable for publication and will be formally accepted for publication once it meets all outstanding technical requirements.

An invoice will be generated when your article is formally accepted. Please note, if your institution has a publishing partnership with PLOS and your article meets the relevant criteria, all or part of your publication costs will be covered. Please make sure your user information is up-to-date by logging into Editorial Manager at Editorial Manager® and clicking the ‘Update My Information' link at the top of the page. For questions related to billing, please contact  and clicking the ‘Update My Information' link at the top of the page. For questions related to billing, please contact billing support..

Kind regards,

Antony Bayer

Academic Editor

PLOS One
---

## [Editor Report · Acceptance letter]

PONE-D-25-50563R2

PLOS One

Dear Dr. Murray,

I'm pleased to inform you that your manuscript has been deemed suitable for publication in PLOS One. Congratulations! Your manuscript is now being handed over to our production team.

Kind regards,

on behalf of

Professor Antony Bayer

Academic Editor

PLOS One